# MemCast: Memory-Driven Time Series Forecasting with Experience-Conditioned Reasoning

Xiaoyu Tao [1]  Mingyue Cheng [1] [*]  Ze Guo [1]  Shuo Yu [1]  Yaguo Liu [1]  Qi Liu [1]  Shijin Wang [2]

## Abstract

Time series forecasting (TSF) plays a critical role in decision-making for many real-world applications. Recently, large language model (LLM)-based forecasters have made promising advancements. Despite their effectiveness, existing methods often lack explicit experience accumulation and continual evolution. In this work, we propose MemCast, a learning-to-memory framework that reformulates TSF as an experience-conditioned reasoning task. Specifically, we learn experience from the training set and organize it into a hierarchical memory. This is achieved by summarizing prediction results into historical patterns, distilling inference trajectories into reasoning wisdom, and inducing extracted temporal features into general laws. Furthermore, during inference, we leverage historical patterns to guide the reasoning process and utilize reasoning wisdom to select better trajectories, while general laws serve as criteria for reflective iteration. Additionally, to enable continual evolution, we design a dynamic confidence adaptation strategy that updates the confidence of individual entries without leaking the test set distribution. Extensive experiments on multiple datasets demonstrate that MemCast consistently outperforms previous methods, validating the effectiveness of our approach. Our code is available at https://github.com/Xiaoyu-Tao/MemCast-TS.

## 1. Introduction

Time series forecasting plays a vital role in decision-making across a wide range of real-world scenarios, including en-

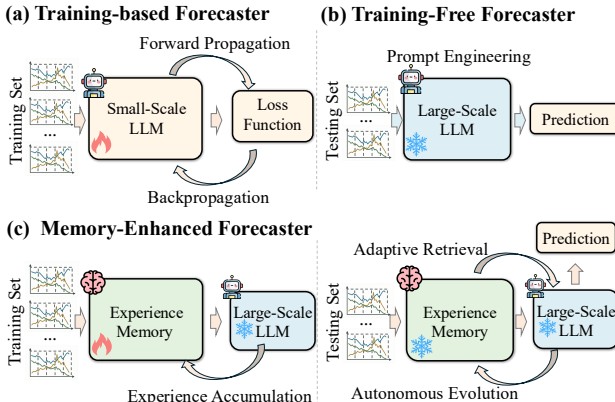

*Figure 1.* Comparison of training-based, training-free, and memory-enhanced LLM forecasting approaches.

ergy scheduling (Qiu et al., 2024), financial trading (Feng et al., 2019), and healthcare monitoring (Huang et al., 2025b). More generally, TSF can be formulated as learning a mapping from historical time series and associated contextual features, including dynamic features that vary over time (e.g., weather information) and static features that remain invariant across the forecasting horizon (e.g., location attributes), to future outcomes (Cheng et al., 2025a).

Building upon this formulation, TSF methods can be broadly categorized into several groups. Early statistical methods predict outcomes under predefined statistical assumptions (Hyndman & Khandakar, 2008). In contrast, data-driven deep learning approaches replace handcrafted statistical assumptions with automatic feature learning (Lin et al., 2025). Recently, LLM-based approaches leverage the capabilities of LLMs for TSF (Huang et al., 2025a). These methods can be divided into two primary categories. As shown in Figure 1 (a), one line relies on backpropagation to update model weights, typically adapting relatively small-scale LLMs to TSF tasks, as represented by Time-LLM (Jin et al., 2024). As shown in Figure 1 (b), the other line adopts prompting strategies that directly harness the reasoning ability of large-scale LLMs, including LSTPrompt (Liu et al., 2024a) and TimeReasoner (Cheng et al., 2026). These approaches offer advantages in capturing temporal dependencies with minimal data dependency.

[*]Corresponding author. [1]State Key Laboratory of Cognitive Intelligence, University of Science and Technology of China, Hefei, China [2]iFLYTEK Research, Hefei, China. Correspondence to: Mingyue Cheng <mycheng@ustc.edu.cn>.

*Proceedings of the 43rd International Conference on Machine Learning*, Seoul, South Korea. PMLR 306, 2026. Copyright 2026 by the author(s).

Despite these advances, most existing approaches still exhibit two limitations. First, they generally treat each forecasting instance as an isolated reasoning process, without explicitly converting historical prediction outcomes, reasoning processes, and temporal regularities across instances into reusable forecasting experience (Zhao et al., 2023). As a result, knowledge acquired from past predictions is difficult to accumulate and transfer to future forecasting instances (Yang et al., 2025). Second, the absence of continual evolution limits the model's potential for lifelong improvement, preventing stored experience from being adaptively refined as new forecasting instances are observed (Chow et al., 2024). As illustrated in Figure 1(c), these limitations motivate a memory-based learning approach for experience-conditioned TSF. Instead of updating LLM parameters through backpropagation, the proposed approach accumulates structured forecasting experience in an external memory on the training set while keeping the LLM parameters frozen, enabling adaptive retrieval and continual memory evolution during inference on the testing set.

Based on the above analysis, we propose MemCast, a learning-to-memory framework that reformulates TSF as an experience-conditioned reasoning task. MemCast begins by extracting forecasting experience from the training set and organizing it into a hierarchical memory. Specifically, prediction outcomes are summarized into historical patterns, inference trajectories are distilled into reusable reasoning wisdom, and statistical features discovered from data are abstracted into general laws. During inference, historical patterns guide the reasoning process, reasoning wisdom supports the selection of effective reasoning trajectories, and general laws provide criteria for reflective iteration. To further support continual evolution without retraining, we introduce a dynamic confidence adaptation strategy that updates entry-level confidence while avoiding test set distribution leakage. Extensive experiments on multiple real-world datasets demonstrate that MemCast consistently outperforms existing methods, validating the effectiveness of learning experience and enabling memory-enhanced forecasting. We hope this work inspires further exploration of experience-driven learning for TSF.

## 2. Related Work

In this section, we review three lines of related work: conventional TSF, recent LLM-based TSF, and memory-augmented reasoning.

### 2.1. Conventional Time Series Forecasting

Time series forecasting has been extensively studied from diverse modeling perspectives. Early research mainly focuses on classical statistical methods, such as ARIMA (Hyndman & Khandakar, 2008) and exponential smoothing (Win-ters, 1960), which model temporal dependencies under predefined assumptions. These methods offer good interpretability, but their capacity is constrained by fixed assumptions and limited flexibility (Han et al., 2025; Zhang et al., 2024). Subsequently, data-driven deep learning methods become dominant by replacing handcrafted assumptions with automatic representation learning. CNN-based models (Cheng et al., 2025b) extract local patterns, RNN-based models (Wang et al., 2019) capture temporal dependencies, GNN-based models (Feng et al., 2019) model relations among interconnected series, and Transformer-based architectures (Shi et al., 2025b) capture long-range interactions. MLP-based approaches (Zeng et al., 2023) further show that simple architectures can achieve competitive performance with high efficiency. These methods improve forecasting accuracy by learning nonlinear dynamics from historical observations. However, they often require large-scale labeled data (Wang et al., 2025) and struggle to incorporate contextual features.

### 2.2. LLM-based Time Series Forecasting

Recently, LLMs have attracted growing interest for TSF due to their contextual modeling and reasoning capabilities (Luo et al., 2025). Existing LLM-based methods leverage encoded knowledge to alleviate data scarcity and enhance contextual understanding (Bian et al., 2024; Shi et al., 2025a). According to whether model parameters are updated, they can be categorized into training-based and training-free approaches (Tang et al., 2025; Jia et al., 2024). Training-based methods adapt LLMs to TSF through supervised fine-tuning, or reinforcement learning (Jin et al., 2024; Tao et al., 2025). These methods introduce task-specific components to bridge the gap between numerical sequences and language-oriented contextual features (Tao et al., 2026). However, due to computational constraints, they are usually applied to relatively small LLMs, limiting reasoning capacity (Liu et al., 2025a; Pan et al., 2024). In contrast, training-free methods employ large-scale LLMs and exploit their reasoning capabilities through prompt design and in-context learning, without parameter updates (Xue & Salim, 2023; Liu et al., 2024a). These methods preserve the general reasoning ability of large LLMs, but their effectiveness often depends on prompts, retrieved examples, or manually designed reasoning instructions. More broadly, both LLM-based forecasting methods usually treat each forecasting instance in isolation (Yang et al., 2025), leaving accumulated forecasting experience insufficiently explored.

### 2.3. Memory-Augmented Reasoning

Inspired by theories of human memory (Gathercole, 1998; Milner et al., 1998), memory mechanisms have been widely used to preserve historical information in neural models. Early sequence models, including RNNs (Elman, 1990),

LSTMs (Graves, 2012), and GRUs (Cho et al., 2014), capture dependencies through recurrent or gated hidden states. Later methods introduce external read-write memory for explicit information storage and access (Sukhbaatar et al., 2015; Graves et al., 2016), while Transformer-XL and Compressive Transformers extend accessible histories through recurrence and compressed memory (Dai et al., 2019; Rae et al., 2020). Recent memory-enhanced LLMs further store, retrieve, and update information beyond the current context, enabling the reuse of historical cases or interaction records for factual grounding, contextual adaptation, and experience-conditioned reasoning (Zhao et al., 2023; Chang et al., 2025). Representative methods include retrieval-based memory, compressed memory, and updatable memory, as shown in MemoRAG (Qian et al., 2025), Memory³ (Yang et al., 2024), and HippoRAG (Gutiérrez et al., 2024). These advances provide a foundation for addressing isolated forecasting in LLM-based TSF by supporting experience accumulation, reasoning strategy reuse, and continual adaptation in non-stationary environments.

## 3. Methodology

In this section, we first formalize the problem definition and then present the overview along with its key components.

### 3.1. Problem Definition

We consider a dataset $\mathcal{D} = \{(X_i, C_i, P_i)\}_{i=1}^N$ consisting of $N$ time series instances. For each instance, $X \in \mathbb{R}^{T \times d}$ denotes the historical time series over $T$ time steps with $d$ dimensions, and $C$ represents the associated contextual features aligned with the historical observations. Specifically, the contextual features include static features $C^s$ and dynamic features $C^d = \{c_1^d, \ldots, c_T^d\}$ aligned with the historical observations. The prediction target is the future time series $P \in \mathbb{R}^{H \times d}$ over a forecasting horizon $H$. Ultimately, the goal of time series forecasting is to learn an underlying mapping function $\mathcal{F}: (X, C^s, C^d) \to P$.

### 3.2. Framework Overview

Figure 2 illustrates the overall framework of MemCast. In the experience memory accumulation phase, the framework learns from the prediction outcomes, inference trajectories, and extracted temporal features on the training set to build a hierarchical memory. In the experience memory utilization phase, MemCast leverages the constructed hierarchical memory to support history-enhanced reasoning, wisdom-driven trajectory exploration, and law-based reflection, thereby enabling experience-conditioned reasoning without retraining parameters. Furthermore, a dynamic confidence adaptation strategy is employed to update the confidence of individual entries for experience memory evolution. The following subsections detail each component.

### 3.3. Experience Memory Encoding

A key question in experience-conditioned forecasting is how to represent historical experience in a form that LLMs can effectively understand, retrieve, and reuse. Existing memory systems often rely on dense vectors (Lewis et al., 2020), key-value memory (Miller et al., 2016), or semi-structured files (Park et al., 2023). While these representations support efficient storage and similarity search, they primarily focus on what to retrieve, rather than why a previous experience is useful. Instead, we adopt a simple yet effective textual encoding strategy. Each historical forecasting case is encoded as a textual memory entry, which records the historical pattern, reasoning wisdom, and general law in a structured natural-language format. This representation is directly compatible with LLMs and can be inserted into the context without additional memory encoders. It also preserves interpretable reasoning traces and supports flexible updates with new observations.

### 3.4. Experience Memory Accumulation

**Historical Pattern Abstraction.** Instance-level historical patterns provide precise references for reasoning when encountering rare scenarios, particularly under distributional shifts (Cheng et al., 2026). While such specific cases are valuable, directly feeding raw numerical values into LLMs is often ineffective for reasoning, as LLMs lack inherent sensitivity to numerical magnitudes and may fail to capture underlying temporal trends (Zhao et al., 2023). Building on the above analysis, we adopt a direct yet effective abstraction strategy that converts raw data into compact semantic summaries. Specifically, for each historical instance consisting of a lookback window $\mathbf{x} \in \mathbb{R}^L$ and a ground truth prediction window $\mathbf{y} \in \mathbb{R}^H$, we employ a summarization LLM $\mathcal{S}$ to translate their numerical dynamics into descriptive natural language text. The final stored memory is structured as a paired summary $\mathcal{M}_{his} = (\mathbf{x}, \mathbf{m}_{out})$, defined as: $\mathbf{m}_{out} = \mathcal{S}(\mathbf{y})$, where $\mathbf{m}_{out}$ encapsulates the trend evolution, volatility, and peak values, respectively.

**Reasoning Wisdom Distillation.** Reasoning trajectories connect raw observations to final predictions and make intermediate decisions interpretable. However, they are usually generated per instance and discarded after inference, limiting cross-sample reuse. Directly storing historical trajectories is impractical due to redundancy and noise in raw reasoning outputs (Yu et al., 2026). To address this, we propose reasoning wisdom distillation to extract reusable experience from instance-level trajectories. Specifically, given a set of generated reasoning trajectories $\mathcal{C} = \{\mathbf{c}_i\}_{i=1}^N$ and their corresponding prediction errors $e_i = \mathcal{L}(\hat{\mathbf{y}}(\mathbf{c}_i), \mathbf{y}_i)$, we partition the trajectories into successful cases $\mathcal{C}_{pos} = \{\mathbf{c}_i \mid e_i < \tau\}$ and failed cases $\mathcal{C}_{neg} = \{\mathbf{c}_i \mid e_i \geq \tau\}$ based on a performance threshold $\tau$.

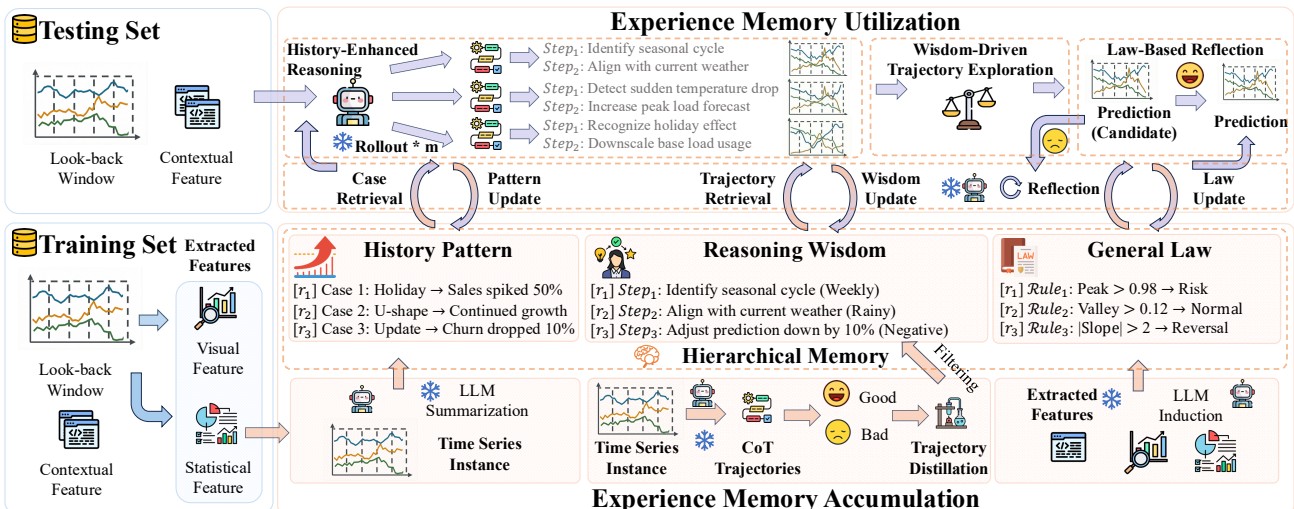

*Figure 2.* Overview of MemCast as an LLM-driven time series forecasting framework that constructs hierarchical memory from the training set via experience accumulation for experience-conditioned reasoning on the testing set.

By analyzing these two groups separately, we distill success wisdom $\mathcal{W}_{pos}$ and failure wisdom $\mathcal{W}_{neg}$, which together constitute the memory module $\mathcal{M}_{rea} = \{\mathcal{W}_{pos}, \mathcal{W}_{neg}\}$. We further implement a filtering to maintain a compact wisdom set, where similarity is computed as a combination of feature-level cosine similarity and raw space dynamic time warping (DTW). The composite similarity score $S(\mathbf{x}_q, \mathbf{x}_k)$ between a query sequence $\mathbf{x}_q$ and a candidate sequence $\mathbf{x}_k$ is defined as follows:

$$S(\mathbf{x}_q, \mathbf{x}_k) = \alpha S_{\text{sem}} + (1 - \alpha)S_{\text{str}}, \tag{1}$$

where the semantic similarity is computed as $S_{\text{sem}} = \text{CosSim}(f(\mathbf{x}_q), f(\mathbf{x}_k))$, and the structural proximity is measured by $S_{\text{str}} = \exp\left(-\text{DTW}(\mathbf{x}_q, \mathbf{x}_k)/\gamma\right)$. Here, $f(\cdot)$ denotes the feature embedding function, $\alpha \in [0, 1]$ is a weighting coefficient, and $\gamma$ is a scaling factor that controls the sensitivity of the DTW-based proximity. Based on the combined score $S$, we replace redundant matches with $S > 0.95$, merge overlapping cases with $0.8 < S \leq 0.95$ via LLM-driven fusion, and preserve novel patterns with $S \leq 0.8$.

**General Law Induction.** General laws aim to prevent the model from deviating from basic physical commonsense when adapting to new instances. While such laws could in principle be derived from instance-level analysis, the large scale of training data and the low information density of raw time series make this approach inefficient (Liu et al., 2025b). Building on this observation, we adopt a feature-level knowledge discovery strategy to induce general laws. Feature representations reduce redundancy, yet manual large-scale analysis remains impractical. We therefore leverage the general capabilities of LLMs as automated law inducers to analyze extracted temporal features and summarize general laws. Specifically, given a training dataset $\mathcal{D} = \{\mathbf{x}_i\}_{i=1}^N$

consisting of $N$ raw time series samples, where $\mathbf{x}_i \in \mathbb{R}^T$. We design a toolset to automatically extract informative features from these samples. Moreover, we employ a textualization module that converts the numerical features of each sample into a textual description $\mathbf{s}_i$, enabling effective processing by LLMs. Based on a collection of these textualized representations, the LLM $\mathcal{S}$ induces general laws that capture common principles: $\mathcal{M}_{gen} = \mathcal{S}(\{\mathbf{s}_1, \ldots, \mathbf{s}_k\})$, where $\{\mathbf{s}_1, \ldots, \mathbf{s}_k\}$ represents a cluster of representative samples derived from $\mathcal{D}$. Finally, the induced general laws $\mathcal{M}_{gen}$ are used as self-reflection criteria during the inference.

### 3.5. Experience Memory Utilization

**History-Enhanced Reasoning.** Existing LLM-based forecasting methods usually treat each forecasting instance in isolation and rely mainly on static parametric knowledge from pre-training. This paradigm underuses recurring empirical patterns in historical data, which are crucial for inferring future trend evolution. Although prior methods retrieve historical examples as demonstrations (Yang et al., 2025), they seldom abstract their temporal evolution into a reusable forecasting experience. Moreover, raw numerical values are not well-suited for LLM reasoning, as LLMs may be insensitive to numerical magnitudes and trend changes. Motivated by this observation, we introduce history-enhanced reasoning based on historical pattern abstraction. To ensure retrieval consistency across memory components, we use the same similarity formulation defined in Eq. (1). Given a query forecasting instance, this process retrieves the top-$k$ most relevant historical patterns from the memory, denoted as $\mathcal{C}_{sim} \subset \mathcal{M}_{his}$. These retrieved entries serve as concrete analogical references that summarize how similar historical instances evolve. We then integrate $\mathcal{C}_{sim}$ with the current

instance to construct an augmented context for the LLM. Consequently, the inference process is transformed from a generic generation probability $P(Y|X)$ into an experience-conditioned probability $P(Y|X, \mathcal{C}_{sim})$. In this way, the model grounds its reasoning in similar historical patterns.

**Wisdom-Driven Trajectory Exploration.** Despite the availability of training-derived experience, inference with LLMs remains uncertain. Different reasoning trajectories may yield inconsistent predictions under similar inputs, especially when retrieved evidence is ambiguous or the future trend is difficult to infer. Relying on a single reasoning pass risks amplifying stochastic errors and undermines reliability. We thus design an uncertainty-aware trajectory exploration strategy that generates multiple candidate trajectories and selects the relatively reliable one with memory guidance. For a query instance $\mathbf{x}$, we first compute a composite similarity score $S$, as defined in Eq. (1), for each candidate experience $\mathcal{E}_j$ stored in the memory $\mathcal{M}_{rea}$. Based on these scores, we retrieve the top-$k$ most relevant experiences to construct the augmented context set $\mathcal{C}_{ret}$. These retrieved reasoning cases provide validated references for which inference trajectories are more likely to yield accurate forecasts. Then, the LLM samples $M$ independent reasoning paths $\{(\mathcal{T}_m, \hat{\mathbf{y}}_m)\}_{m=1}^{M}$, where $\mathcal{T}_m$ represents the generated chain-of-thought rationale and $\hat{\mathbf{y}}_m$ denotes the corresponding forecast values. To select the final output, we apply a scoring function $\phi(\cdot)$ that measures the semantic consistency of each trajectory $\mathcal{T}_m$ against the validated reasoning wisdom $\mathcal{C}_{ret}$. The candidate prediction $\hat{\mathbf{y}}$ is assigned as $\hat{\mathbf{y}}_{m^*}$, where $m^*$ denotes the trajectory with the highest reliability score: $m^* = \arg\max_m \phi(\mathcal{T}_m), \quad \hat{\mathbf{y}} = \hat{\mathbf{y}}_{m^*}$.

**Law-Based Reflection.** To mitigate the risk that predicted results may be inconsistent with real-world constraints, we design a law-based reflection strategy. Specifically, after the model generates a candidate prediction $\hat{\mathbf{y}}$, we evaluate the predicted sequence against a pre-constructed set of domain-specific general laws $\mathcal{M}_{gen} = \{r_1, r_2, \ldots, r_K\}$. These laws encapsulate essential constraints, such as non-negativity or range-bound continuity checks, and provide explicit criteria for verifying prediction feasibility. By externalizing such constraints as general laws, the model can conduct post-hoc self-correction in a more controllable and interpretable manner. Formally, if there exists a rule $r_k \in \mathcal{M}_{gen}$ such that the constraint $r_k(\hat{\mathbf{y}})$ is not satisfied, an immediate re-reasoning loop is triggered. Unlike simple rejection sampling, this loop feeds the specific violation details back into the LLM as a corrective prompt, forcing the model to revise its prediction with awareness of the violated constraint (Shinn et al., 2023). This iterative refinement continues until all constraints are met or a maximum retry limit is reached, ensuring that the final output $Y$ is not only statistically plausible but also compliant with domain knowledge.

*Table 1.* Overview of diverse real-world datasets with rich contextual features. These benchmarks span multiple domains and temporal frequencies to ensure comprehensive evaluation.

| Dataset | Domain | Length | Variables | Frequency |
|---------|--------|--------|-----------|-----------|
| NP | Energy | 14,496 | 3 | 1 hour |
| PJM | Energy | 14,496 | 3 | 1 hour |
| BE | Energy | 14,496 | 3 | 1 hour |
| FR | Energy | 14,496 | 3 | 1 hour |
| DE | Energy | 14,496 | 3 | 1 hour |
| ETTh | Electricity | 12,240 | 7 | 1 hour |
| ETTm | Electricity | 24,096 | 7 | 15 mins |
| WP | Energy | 24,096 | 7 | 15 mins |
| SP | Energy | 24,096 | 7 | 15 mins |
| MOPEX | Environment | 12,240 | 6 | 1 day |

### 3.6. Experience Memory Evolution

Memory can be either static or dynamic. Static memory keeps historical experience unchanged, which is less suitable for TSF, where temporal patterns and contextual features continuously evolve. Thus, the usefulness of historical experience should not be fixed. We adopt a dynamic memory design where reliability evolves with forecasting feedback, while memory content remains unchanged to avoid test distribution leakage. Specifically, hierarchical experience learning accumulates structured experience from the training set. However, modifying experience content during inference may bias memory toward the test distribution, undermining fair evaluation and generalization. To address this, we introduce a confidence-aware refinement strategy that decouples experience accumulation from utilization. Each training-derived experience is assigned a confidence score reflecting its empirical reliability and contribution to successful predictions. During inference, relevant experiences are retrieved by semantic similarity. Once the ground truth becomes available, we compare the generated forecast $\hat{\mathbf{y}}$ with $\mathbf{y}$ using a moving average (MA) baseline, avoiding updates based solely on absolute error. A prediction is successful if $\mathcal{L}_{LLM} < \mathcal{L}_{MA}$. Upon success, the confidence scores of contributing experiences are increased. Otherwise, no update is performed. Since only confidence weights are updated, this strategy enables continual memory evolution while preserving train–test separation.

## 4. Experiments

In this section, we first introduce the experimental settings and then present experimental results, ablation studies, and case analyses to evaluate the proposed framework.

### 4.1. Experimental Settings

**Datasets.** Table 1 provides detailed descriptions of the datasets used in our experiments, covering diverse forecasting scenarios and temporal scales with rich contextual

*Table 2.* Overall forecasting performance under short-term and long-term forecasting on various benchmark datasets. Lower values indicate better performance. The best results are highlighted in **bold**, and the second-best are underlined.

| Model | NP | | PJM | | BE | | FR | | DE | | ETTh | | ETTm | | WP | | SP | | MOPEX | |
|---|---|---|---|---|---|---|---|---|---|---|---|---|---|---|---|---|---|---|---|---|
| | MSE | MAE | MSE | MAE | MSE | MAE | MSE | MAE | MSE | MAE | MSE | MAE | MSE | MAE | MSE | MAE | MSE | MAE | MSE | MAE |
| ARIMA | 78.944 | 5.722 | 181.141 | 9.944 | 6,813.559 | 20.889 | 3,072.651 | 16.878 | 414.126 | 14.568 | 8.992 | 2.314 | 4.456 | 1.725 | 2,504.541 | 32.558 | 120.801 | 6.595 | 7.703 | 1.680 |
| Prophet | 58.424 | 5.311 | 63.342 | 6.102 | 1,055.918 | 18.334 | 999.577 | 14.887 | 515.070 | 15.759 | 27.715 | 3.582 | 22.517 | 3.347 | 6,529.269 | 53.022 | 204.919 | 11.457 | 17.611 | 2.745 |
| PatchTST | 59.305 | 5.326 | 124.575 | 8.686 | 1,120.463 | 18.449 | 1,339.507 | 15.768 | 290.971 | 11.974 | 8.212 | 2.162 | 4.139 | 1.497 | **2,163.256** | 31.822 | 25.523 | 3.612 | 6.506 | 1.736 |
| iTransformer | 64.080 | 5.622 | 116.768 | 8.455 | 1,236.950 | 19.406 | 1,374.755 | 17.980 | 345.994 | 13.304 | 8.780 | 2.277 | 4.152 | 1.505 | 2,218.198 | 32.001 | 26.811 | 3.751 | 7.059 | 1.821 |
| TimeXer | 59.529 | 5.220 | 84.743 | 6.918 | 887.125 | 14.463 | 876.179 | **10.686** | 296.232 | 11.827 | 7.913 | 2.137 | 4.026 | 1.463 | 2,654.421 | 35.840 | 27.792 | 3.702 | 6.402 | 1.693 |
| ConvTimeNet | 55.857 | 5.111 | 108.312 | 7.935 | 1,146.507 | 18.191 | 1,175.164 | 15.096 | 327.377 | 12.727 | 9.101 | 2.356 | 4.346 | 1.593 | 2,218.109 | 32.496 | 27.381 | 3.802 | 7.102 | 1.831 |
| DLinear | 68.723 | 5.798 | 122.269 | 8.800 | 1,074.966 | 18.459 | 1,134.095 | 16.237 | 442.229 | 15.299 | 9.817 | 2.465 | 4.432 | 1.674 | 2,253.934 | 32.509 | 23.720 | 3.506 | 6.752 | 1.793 |
| LSTPrompt | 53.062 | 4.933 | 107.623 | 7.580 | 1,027.130 | 17.379 | 1,105.157 | 14.419 | 393.245 | 13.831 | 14.462 | 2.815 | 8.246 | 1.873 | 3,126.894 | 36.204 | 136.214 | 6.842 | 8.703 | 1.692 |
| LLM-Time | 92.740 | 6.446 | 158.170 | 9.350 | 1,283.180 | 20.078 | 1,282.570 | 17.935 | 379.850 | 14.137 | 27.387 | 3.316 | 5.398 | 1.811 | 4,438.044 | 38.913 | 131.182 | 5.617 | 6.729 | **1.497** |
| TimeReasoner | 70.514 | 5.447 | 56.471 | 5.168 | 780.831 | 12.877 | 837.409 | 11.581 | 306.511 | 12.706 | 10.357 | 2.431 | 4.038 | 1.574 | 2,428.241 | 33.571 | 27.551 | 2.350 | 7.177 | 1.568 |
| Time-LLM | 78.258 | 6.172 | 141.920 | 9.226 | 1,349.527 | 21.656 | 1,419.558 | 18.875 | 533.419 | 16.498 | 11.317 | 2.679 | 4.687 | 1.851 | 2,769.212 | 36.764 | 68.446 | 6.249 | 6.274 | 1.692 |
| **MemCast** | **36.871** | **3.567** | **30.238** | **3.575** | **756.308** | **12.534** | **756.981** | 12.142 | **210.445** | **9.195** | **6.551** | **1.968** | **3.999** | **1.441** | 2,170.456 | **31.715** | **22.998** | **1.876** | **6.193** | 1.525 |

features. Among them, NP, PJM, BE, FR, and DE from the electricity price forecasting (EPF) benchmark (Lago et al., 2021) provide short-horizon electricity price series from different regional power markets, together with corresponding contextual information such as load and generation forecasting. For long-term forecasting, ETTh and ETTm from the ETT benchmark (Zhou et al., 2021) mainly record transformer temperature and load measurements at different sampling frequencies. In addition, Windy Power (WP) and Sunny Power (SP) (iFLYTEK AI Challenge, 2025) contain renewable energy generation data accompanied by meteorological variables, while MOPEX (Makovoz & Marleau, 2005) collects hydrological streamflow with associated meteorological contextual features. These datasets exhibit diverse temporal dynamics and contextual dependencies, forming a comprehensive evaluation. Detailed dataset descriptions are provided in the Appendix A.1.

**Baselines.** We compare MemCast against a diverse set of representative baselines, grouped into four categories for comprehensive evaluation. For statistical methods, we include ARIMA (Hyndman & Khandakar, 2008) and Prophet (Taylor & Letham, 2018), which model temporal patterns based on classical assumptions and trend decomposition. For deep learning-based approaches, we evaluate PatchTST (Nie et al., 2023), iTransformer (Liu et al., 2024b), TimeXer (Wang et al., 2024), ConvTimeNet (Cheng et al., 2025b), and DLinear (Zeng et al., 2023), which leverage advanced neural architectures such as Transformers, CNNs, and MLP models to effectively capture temporal dependencies. In the LLM-based forecasting category, we consider LSTPrompt (Liu et al., 2024a), LLM-Time (Gruver et al., 2023), TimeReasoner, and Time-LLM (Jin et al., 2024), which adapt LLMs to TSF through prompting, reasoning, and alignment mechanisms. These baselines cover both conventional forecasting paradigms and recent LLM-driven methods, enabling a comprehensive comparison across model assumptions and reasoning capabilities. Further implementation details are provided in Appendix A.2.

**Implementation Details.** In this work, we use GPT-5 (Singh et al., 2025) as the reasoning engine for all LLM-based inference. By default, all experiments are conducted via the official API, with the temperature set to 0.6, top-p to 0.7, and the maximum number of generated tokens set to 16,384. Deep learning baselines are trained in PyTorch using the Adam optimizer on a single NVIDIA GeForce RTX 4090D GPU, following official implementations and recommended hyperparameter configurations. For long-term forecasting tasks, both the look-back window and the prediction horizon are set to 96 time steps, while for short-term forecasting tasks, the look-back window is set to 168 and the prediction horizon to 24. Mean squared error (MSE) and mean absolute error (MAE) are adopted as evaluation metrics. For fair comparison, all models are evaluated on raw time series without normalization, so as to preserve the physical meaning of the original numerical values. The memory is not allowed to grow indefinitely. In practice, memory is maintained with a bounded size through filtering and updating mechanisms, while dynamic confidence adaptation updates the weights of existing entries.

### 4.2. Main Results

Table 2 summarizes the performance on diverse context-rich benchmarks. Overall, MemCast achieves the best results on most metrics, surpassing different baselines in a broad range of forecasting scenarios. Specifically, MemCast demonstrates consistently stronger performance than representative deep learning models across multiple datasets, with particularly noticeable improvements on highly volatile benchmarks such as NP and PJM. These results suggest that reasoning based on historical patterns may offer certain advantages in capturing irregular market dynamics, whereas approaches relying primarily on attention mechanisms may face limitations in such settings. Within the class of LLM-based forecasting methods, MemCast achieves lower prediction errors than Time-LLM on complex and noisy datasets such as BE. This observation indicates that, although fine-tuning-based approaches can adapt model behavior through

*Table 3.* Ablation study on the different memory components: historical pattern, reasoning wisdom, and general law. The consistent performance degradation in "w/o" variants confirms the necessity of each module for accurate forecasting.

| Method | NP MSE | NP MAE | PJM MSE | PJM MAE | BE MSE | BE MAE | FR MSE | FR MAE | DE MSE | DE MAE | ETTh MSE | ETTh MAE | ETTm MSE | ETTm MAE | WP MSE | WP MAE | SP MSE | SP MAE | MOPEX MSE | MOPEX MAE |
|---|---|---|---|---|---|---|---|---|---|---|---|---|---|---|---|---|---|---|---|---|
| w/o All | 65.275 | 5.352 | 39.512 | 4.332 | 866.035 | 14.799 | 865.419 | 14.220 | 364.349 | 13.203 | 8.664 | 2.400 | 6.221 | 1.562 | 3079.615 | 36.355 | 25.255 | 2.043 | 15.111 | 1.978 |
| w/o Law | 46.463 | 3.811 | 33.172 | 3.579 | 798.021 | 13.115 | 821.493 | 13.116 | 297.213 | 11.321 | 6.754 | 2.054 | 5.212 | 1.329 | 2712.801 | 35.937 | 23.488 | 1.971 | 13.381 | 1.991 |
| w/o Wisdom | 55.545 | 5.334 | 33.301 | 3.731 | 858.120 | 14.165 | 826.714 | 13.816 | 314.471 | 12.319 | 8.215 | 2.326 | 5.592 | 1.716 | 2994.571 | 37.794 | 23.558 | 1.940 | 7.139 | 1.678 |
| w/o Pattern | 45.594 | 4.573 | 39.259 | 4.083 | 872.001 | 14.376 | 813.821 | 13.444 | 356.390 | 13.742 | 8.093 | 2.231 | 5.763 | 1.789 | 3656.032 | 39.969 | 30.340 | 2.084 | 10.978 | 1.916 |
| **Ours** | **36.871** | **3.567** | **30.238** | **3.575** | **756.308** | **12.534** | **756.981** | **12.142** | **210.445** | **9.195** | **6.551** | **1.968** | **3.999** | **1.441** | **2,170.456** | **31.715** | **22.998** | **1.876** | **6.193** | **1.525** |

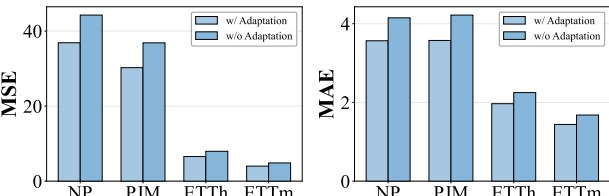

*Figure 3.* Ablation on dynamic confidence adaptation. Dynamic confidence adaptation leads to consistently lower errors compared with the variant without adaptation.

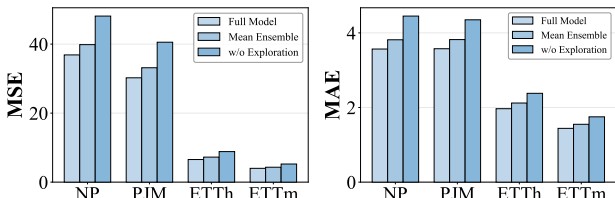

*Figure 4.* Exploration on aggregation strategies. The full model outperforms baselines, confirming that active selection via semantic consistency surpasses passive aggregation.

parameter updates, their generalization ability may be challenged in high-noise scenarios. Furthermore, compared with TimeReasoner, MemCast attains more stable performance across most benchmarks, while TimeReasoner remains competitive on several metrics. The overall improvement is likely related to the explicit experience accumulation in MemCast, which allows historical forecasting experience to be reused across instances rather than being discarded after individual reasoning processes. Overall, these results empirically validate reformulating time series forecasting as experience-conditioned reasoning.

### 4.3. Ablation Studies

**Ablation on Hierarchical Memory.** Table 3 analyzes how memory components affect forecasting performance. Incorporating memory improves over the memory-free baseline. We decompose experience memory into three components: general laws, which impose high-level distributional constraints; reasoning wisdom, which verifies the logical consistency of the prediction process; and historical patterns, which capture instance-level evolutionary dynamics. While most variants degrade relative to the full model, distinct domain dependencies emerge. For instance, on PJM, removing patterns causes the largest degradation, suggesting the importance of instance-level analogical references. For ETTh, removing reasoning wisdom or historical patterns leads to larger errors than removing general laws, indicating the importance of trajectory-level validation and case-level cues. These complementary effects suggest that memories provide non-redundant forecasting guidance. Overall, the superior performance of the full model demonstrates its ability to integrate global constraints, logical verification, and instance cues for experience-conditioned forecasting.

**Ablation on Dynamic Confidence Adaptation.** Figure 3 presents the ablation results on dynamic confidence adaptation across multiple datasets. The variant with dynamic confidence adaptation consistently achieves lower MSE and MAE than the one without adaptation, with particularly noticeable improvements on the highly volatile NP and PJM datasets. This may indicate that dynamically updating confidence helps stabilize the reasoning process when facing rapidly changing conditions. On long-term benchmarks such as ETTh and ETTm, dynamic confidence adaptation also leads to consistent improvements. These observations suggest that, beyond highly volatile scenarios, confidence updating remains beneficial for maintaining reliable performance over extended forecasting horizons. Overall, the results indicate that enabling the memory to continuously evolve during testing contributes to more stable performance. This further confirms that adaptively calibrating the reliability of memory entries is important for reusing experience under changing temporal patterns.

### 4.4. Exploration Analysis

**Exploration on Aggregation Strategy.** Figure 4 illustrates the effectiveness of our uncertainty-aware trajectory exploration strategy. The experiment compares our approach against a baseline without refinement, which relies on a single stochastic pass and a mean ensemble method that averages predictions across $M$ sampled paths. While the mean ensemble yields marginal gains by smoothing out variance, it indiscriminately incorporates both high-quality rationales and inconsistent predictions, thereby diluting accuracy. In contrast, our full model leverages the scoring function $\phi(\cdot)$ to measure the semantic consistency of each generated trajectory $\mathcal{T}_m$ against the retrieved reasoning wis-

*Table 4.* Exploration on different LLM backbones. We evaluate the scalability of our framework across different LLMs.

| Backbone | NP | | PJM | | ETTh | |
|---|---|---|---|---|---|---|
| | MSE | MAE | MSE | MAE | MSE | MAE |
| GPT-4o-mini | 38.075 | 3.957 | 66.333 | 6.159 | 34.070 | 4.114 |
| Gemini 2.5 Pro | 61.822 | 5.149 | 41.077 | 4.121 | 6.726 | 2.030 |
| DeepSeek-V3.1 | 41.707 | **3.539** | 145.995 | 5.021 | 27.066 | 2.633 |
| DeepSeek-R1 | 55.058 | 5.107 | 52.613 | 4.928 | 7.140 | 2.031 |
| **Ours** | **36.871** | 3.567 | **30.238** | **3.575** | **6.551** | **1.968** |

*Table 5.* Exploration of contextual features. Integrating both static and dynamic features generally improves forecasting performance.

| Method | NP | | PJM | | ETTh | |
|---|---|---|---|---|---|---|
| | MSE | MAE | MSE | MAE | MSE | MAE |
| w/o All | 41.133 | **3.403** | 42.315 | 4.469 | 8.041 | 2.249 |
| w/o Dynamic | 37.800 | 3.546 | 37.374 | 4.411 | 8.185 | 2.246 |
| w/o Static | 37.697 | 3.450 | 32.079 | 3.818 | 7.234 | 2.167 |
| **Ours** | **36.871** | 3.567 | **30.238** | **3.575** | **6.551** | **1.968** |

*Table 6.* Sensitivity analysis on the number of retrieved cases ($k$). Top-$k = 3$ consistently achieves the best performance.

| Retrieval Number | NP | | PJM | | ETTh | |
|---|---|---|---|---|---|---|
| | MSE | MAE | MSE | MAE | MSE | MAE |
| Top-$k = 1$ | 37.347 | 3.657 | 37.120 | 3.902 | 7.808 | 2.207 |
| Top-$k = 3$ | **36.871** | **3.567** | **30.238** | **3.575** | **6.551** | **1.968** |
| Top-$k = 5$ | 43.095 | 3.979 | 40.643 | 4.092 | 6.691 | 2.155 |
| Top-$k = 7$ | 46.155 | 4.134 | 45.221 | 3.865 | 7.362 | 2.207 |

dom. By executing this generate-then-select protocol and specifically identifying the trajectory $\mathcal{T}_{m^*}$ that maximizes the reliability score, we effectively filter out stochastic noise. This suggests that active selection grounded in experience-conditioned consistency tends to outperform passive aggregation by better emphasizing reliable trajectories.

**Exploration on LLM Backbone.** Table 4 reports the scalability analysis of our framework with different LLM backbones. As shown in the table, forecasting performance varies across backbones, while stronger reasoning models often achieve competitive results on several datasets. This trend shows that our framework can effectively leverage the backbone's reasoning capacity rather than relying on model-specific heuristics. Moreover, the consistent applicability across different backbones indicates that the learning-to-memory design is backbone-agnostic and transferable to LLMs with varying capacities. Stronger backbones may better interpret retrieved experience, select plausible reasoning trajectories, and conduct reflective refinement, thereby improving the utilization of hierarchical memory. These results suggest that our framework can benefit from future advances in LLM reasoning while maintaining a unified memory-based forecasting paradigm. Importantly, the default configuration achieves strong overall performance, indicating a favorable balance between model capacity and reasoning effectiveness.

**Exploration on Contextual Feature.** Table 5 reports the ablation results of textual components. Overall, the full model achieves the lowest MSE on all datasets and the best results on both metrics for PJM and ETTh, showing that contextual features generally improve forecasting reliability. The only exception is NP MAE, where removing textual inputs slightly improves performance, suggesting

that context may introduce noise in highly volatile market scenarios. Comparing partial variants, removing dynamic context usually leads to a larger performance drop than removing static context, especially on PJM and ETTh, indicating the importance of time-varying cues for capturing evolving conditions. Static context still provides complementary background knowledge. These results show that static and dynamic contexts are mutually beneficial, with dataset-dependent contributions.

### 4.5. Hyperparameter Sensitivity Analysis

**Impact of Retrieval Volume.** Table 6 presents a sensitivity analysis on the number of retrieved cases. Results indicate that a moderate number of cases yields the most accurate performance, effectively balancing useful inductive bias against noise. When retrieval is minimal, the model struggles to abstract reliable commonalities, hindering generalization. Conversely, excessive retrieval degrades performance, as less relevant instances dilute the reasoning focus and introduce interference. This suggests that retrieval quality is more important than merely increasing the amount of retrieved experience. Furthermore, while long-term dependency tasks benefit from richer context, the moderate setting remains the most stable default.

**Impact of Sampled Trajectory.** As illustrated in Figure 6 (a), the forecasting error initially exhibits a significant downward trend as the number of sampled trajectories increases across all datasets. This suggests that generating multiple independent reasoning paths allows the model to explore a broader solution space and effectively mitigate individual hallucinations through consistency verification. However, performance gains tend to saturate or slightly degrade when the number of trajectories reaches a higher level (e.g., $m > 4$). This saturation indicates that while adequate sampling diversity is crucial for stability, an excessive number of paths may introduce lower-quality or redundant traces that dilute the final aggregation. Consequently, an intermediate setting provides a cost-effective balance between accuracy and computational efficiency.

**Impact of Sampling Strategy.** As shown in Figure 6 (b), the results regarding sampling randomness exhibit a distinct U-shaped performance curve. A moderate temperature

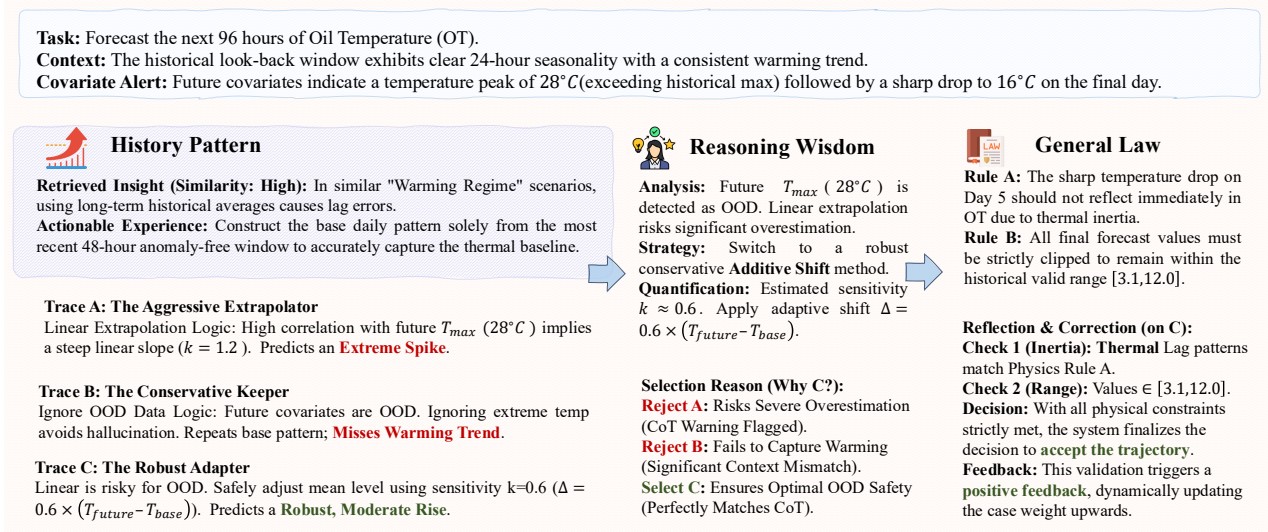

*Figure 5.* A detailed case study on the oil temperature (OT) forecasting task, illustrating experience-conditioned reasoning via constructed memory to effectively handle out-of-distribution thermal shifts.

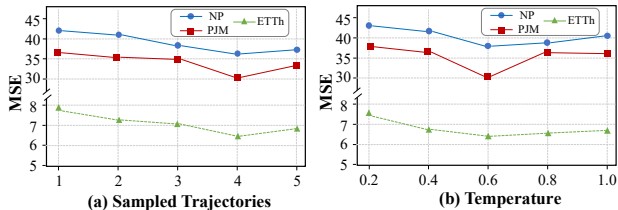

*Figure 6.* Hyperparameter sensitivity analysis regarding the number of sampled trajectories (a) and sampling temperature (b), evaluated across representative benchmark datasets.

consistently yields the most favorable performance. At higher temperatures, the accuracy deteriorates, likely due to the injection of excessive randomness leading to unstable reasoning and hallucinations inconsistent with temporal constraints. Conversely, an extremely low temperature also results in suboptimal performance, suggesting that overly deterministic decoding limits the diversity of the model and its ability to adapt to complex, non-stationary patterns. Therefore, selecting a moderate temperature effectively balances the need for precise, rigorous reasoning with sufficient diversity to capture potential regime shifts.

### 4.6. Case Study

To intuitively demonstrate how MemCast mitigates reasoning hallucinations under distribution shifts, we analyze an oil temperature (OT) forecasting scenario where future covariates exhibit an anomalous out-of-distribution (OOD) peak of 28°C. Faced with this "Warming Regime," the history pattern memory first retrieves actionable experience to construct a baseline solely from the recent 48-hour window, thereby avoiding lag errors associated with long-term aver-

ages. Subsequently, the reasoning wisdom acts as a logical filter, scrutinizing generated trajectories to reject both the aggressive linear extrapolation of Trajectory A ($k = 1.2$) and the overly conservative stagnation of Trajectory B. Instead, it selects the effective adapter (Trajectory C), which applies a moderate sensitivity ($k = 0.6$) to balance the covariate's influence with historical inertia. Finally, the prediction is validated by the general law, ensuring compliance with physical constraints such as the valid range of $[3.1, 12.0]$. This process highlights MemCast's ability to actively reason through physical constraints rather than blindly following misleading OOD signals.

## 5. Conclusion

In this work, we presented MemCast, a learning-to-memory framework that reformulates TSF as an experience-conditioned reasoning problem. By accumulating forecasting experience from the training set, MemCast constructs a hierarchical memory composed of historical patterns, reasoning wisdom, and general laws, enabling the model to move beyond instance-level reasoning. During inference, these complementary memory components collaboratively guide reasoning, trajectory selection, and reflective correction, while a dynamic confidence adaptation strategy supports continual evolution without introducing test-data leakage. Extensive experiments across diverse datasets demonstrate that MemCast consistently outperforms existing approaches, highlighting the effectiveness of explicit experience abstraction and controlled memory utilization for accurate time series forecasting. We believe that MemCast offers a promising step toward experience-conditioned forecasting with continual evolution.

## Impact Statement

This work advances time series forecasting through Mem-Cast, an experience-conditioned reasoning framework that enhances interpretability and reliability in critical domains such as energy and healthcare. By integrating general laws to enforce physical constraints and employing dynamic confidence adaptation for autonomous evolution, our approach effectively mitigates generative hallucinations and adapts to non-stationary environments.

## Conflict of Interest Disclosure

The authors declare no financial conflicts of interest related to this work.

## Acknowledgements

This work was supported by grants from the National Natural Science Foundation of China (No. 62502486), the Fundamental Research Funds for the Central Universities (No. JZ2025HGTB0240), Guangdong S&T Programme (No. 2025B0101120004), the grants of the Provincial Natural Science Foundation of Anhui Province (No. 2408085QF193), the Fundamental Research Funds for the Central Universities of China (No. WK2150110032), USTC Research Funds of the DoubleFirst-Class Initiative (No. YD2150002501).

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

# A. Appendix

## A.1. Dataset Descriptions

*Table 7.* Dataset descriptions. The dataset size is organized in (Train, Validation, Test).

| Dataset | Exogenous Descriptions | Endogenous Descriptions | Sampling Frequency | Dataset Size |
|---|---|---|---|---|
| NP | Grid Load, Wind Power | Nord Pool Electricity Price | 1 Hour | (10147,1450,2899) |
| PJM | System Load, COMED zonal load | Pennsylvania-New Jersey-Maryland Electricity Price | 1 Hour | (10147,1450,2899) |
| BE | Generation, System Load | Belgium's Electricity Price | 1 Hour | (10147,1450,2899) |
| FR | Generation, System Load | France's Electricity Price | 1 Hour | (10147,1450,2899) |
| DE | Wind power, Amprion zonal load | German Electricity Price | 1 Hour | (10147,1450,2899) |
| ETTh1 | Power Load Feature | Oil Temperature | 1 Hour | (7344,2448,2448) |
| ETTm1 | Power Load Feature | Oil Temperature | 15 Minutes | (14457,4820,4819) |
| WP | Meteorological Variables | Power Generation | 15 Minutes | (14457,4820,4819) |
| SP | Meteorological Variables | Power Generation | 15 Minutes | (14457,4820,4819) |
| MOPEX | Meteorological Forcing Variables | Daily Streamflow Discharge | 1 Day | (7344,2448,2448) |

We conduct a comprehensive evaluation of our method across diverse domains, covering both short-term and long-term forecasting tasks. The benchmarks include electricity prices, power load, renewable energy generation, and hydrological factors, featuring varying sampling frequencies from 15 minutes to 24 hours. See Table 7 for the detailed statistics.

**Short-term Forecasting Benchmarks.** We utilize five standard datasets from the Electricity Price Forecasting (EPF) benchmark[1], all sampled at an hourly frequency. These datasets are multivariate, containing past electricity prices along with distinct exogenous variables (e.g., load and generation forecasts) relevant to their respective markets:

- **Nord Pool (NP):** Sourced from the Nordic electricity market. It includes hourly electricity prices alongside exogenous forecasts for grid load and wind power generation.

- **Pennsylvania-New Jersey-Maryland (PJM):** Represents the US PJM interconnection. It features zonal electricity prices for the Commonwealth Edison (COMED) zone, incorporating system-wide load and zonal load forecasts as covariates.

- **Belgium (BE):** Sourced from the Belgian power exchange. It comprises hourly prices, domestic load forecasts, and cross-border generation forecasts from the French grid.

- **France (FR):** Represents the French electricity market. The dataset pairs hourly prices with corresponding national load forecasts and power generation predictions.

- **Germany (DE):** Sourced from the German market. It includes hourly prices, zonal load forecasts for the Amprion TSO area, and renewable energy forecasts for both wind and solar power generation.

**Long-term Forecasting Benchmarks.** For long-term horizons, we evaluate performance on datasets spanning industrial power transformers, renewable energy telemetry, and hydrological systems:

- **Electricity Transformer Temperature (ETTh1 & ETTm1):** A widely used benchmark for long-term forecasting[2]. These datasets record the Oil Temperature (OT) and six load-type features from electricity transformers. **ETTh1** is sampled at an hourly frequency, while **ETTm1** is sampled every 15 minutes.

- **Wind Power (WP):** Derived from the iFLYTEK AI Developer Competition[3], this dataset records the actual power output from a wind farm at a 15-minute resolution. It incorporates six meteorological covariates: direct radiation, wind speed (80m), wind direction (80m), temperature (2m), humidity (2m), and precipitation.

- **Sunny Power (SP):** Also sourced from the iFLYTEK competition with a 15-minute frequency. It tracks power generation from a separate renewable site and shares the same set of six meteorological features as the WP dataset.

---

[1] https://github.com/jeslago/epftoolbox
[2] https://github.com/zhouhaoyi/ETDataset
[3] https://challenge.xfyun.cn/topic/info?type=renewable-power-forecast&option=ssgy&ch=dwsf259

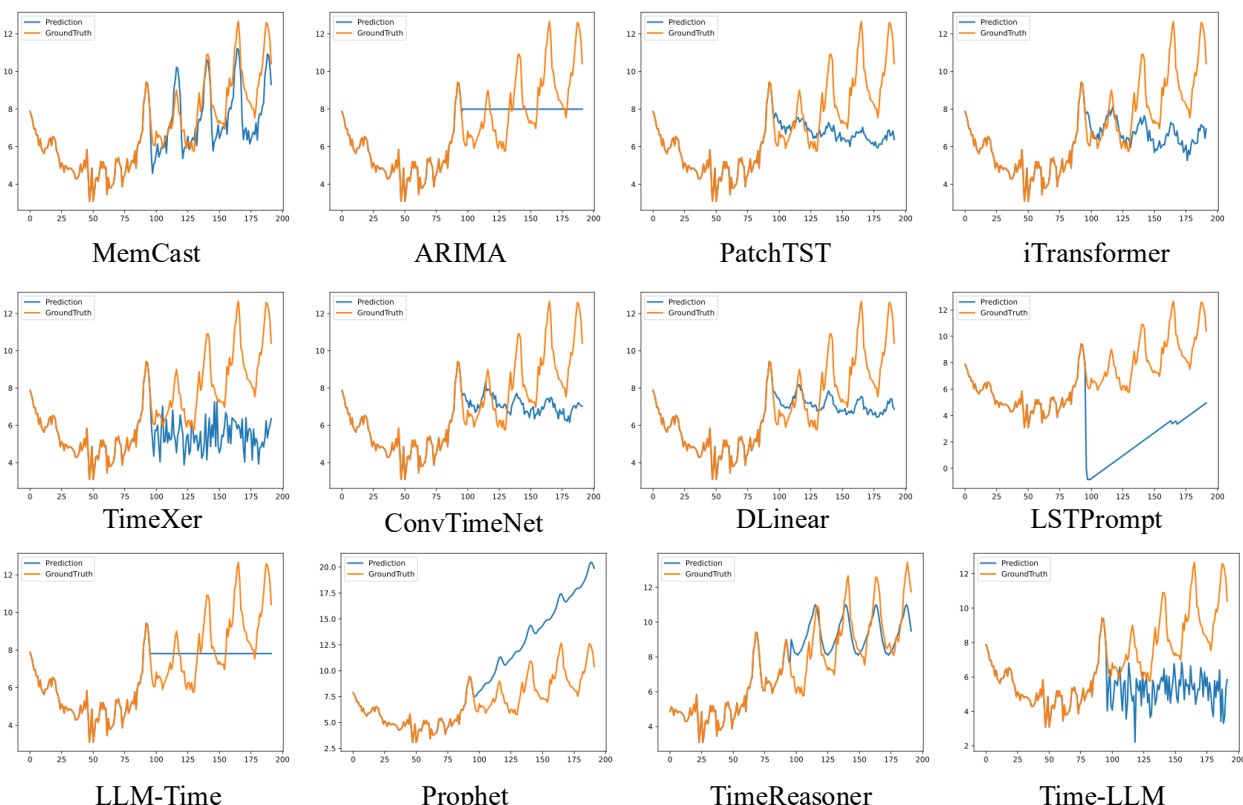

*Figure 7.* Qualitative visualization demonstrates that our framework captures complex non-stationary patterns and sharp fluctuations more accurately than state-of-the-art baselines, which generally suffer from significant lag or amplitude mismatch.

- **Model Parameter Estimation Experiment (MOPEX):** A hydrological dataset widely used for rainfall-runoff modeling[4]. Sampled at a daily frequency, it targets streamflow discharge prediction and includes four meteorological drivers: Mean Areal Precipitation (MAP), Climatic Potential Evaporation (CPE), Daily Maximum Air Temperature ($T_{max}$), and Daily Minimum Air Temperature ($T_{min}$).

## A.2. Compared Baselines

We compare MemCast against a diverse set of representative baselines, ranging from classical statistical methods to state-of-the-art foundation models.

- **ARIMA** (Hyndman & Khandakar, 2008): A classic statistical method that models temporal patterns using autoregression, differencing, and moving averages to capture linear dependencies.

- **Prophet** (Taylor & Letham, 2018): An additive regression model designed for business time series, which effectively decomposes data into trends, seasonality, and holiday effects.

- **DLinear** (Zeng et al., 2023): A simple yet effective MLP-based model that utilizes a decomposition layer to handle trend and seasonal components separately.

- **PatchTST** (Nie et al., 2023): A Transformer-based model that introduces channel independence and patch-based tokenization to capture local semantic information and reduce computational complexity.

- **iTransformer** (Liu et al., 2024b): An inverted Transformer architecture that embeds the whole time series of each variate as a token and applies attention mechanisms across multivariate channels.

---

[4]https://irsa.ipac.caltech.edu/data/SPITZER/docs/dataanalysistools/tools/mopex/

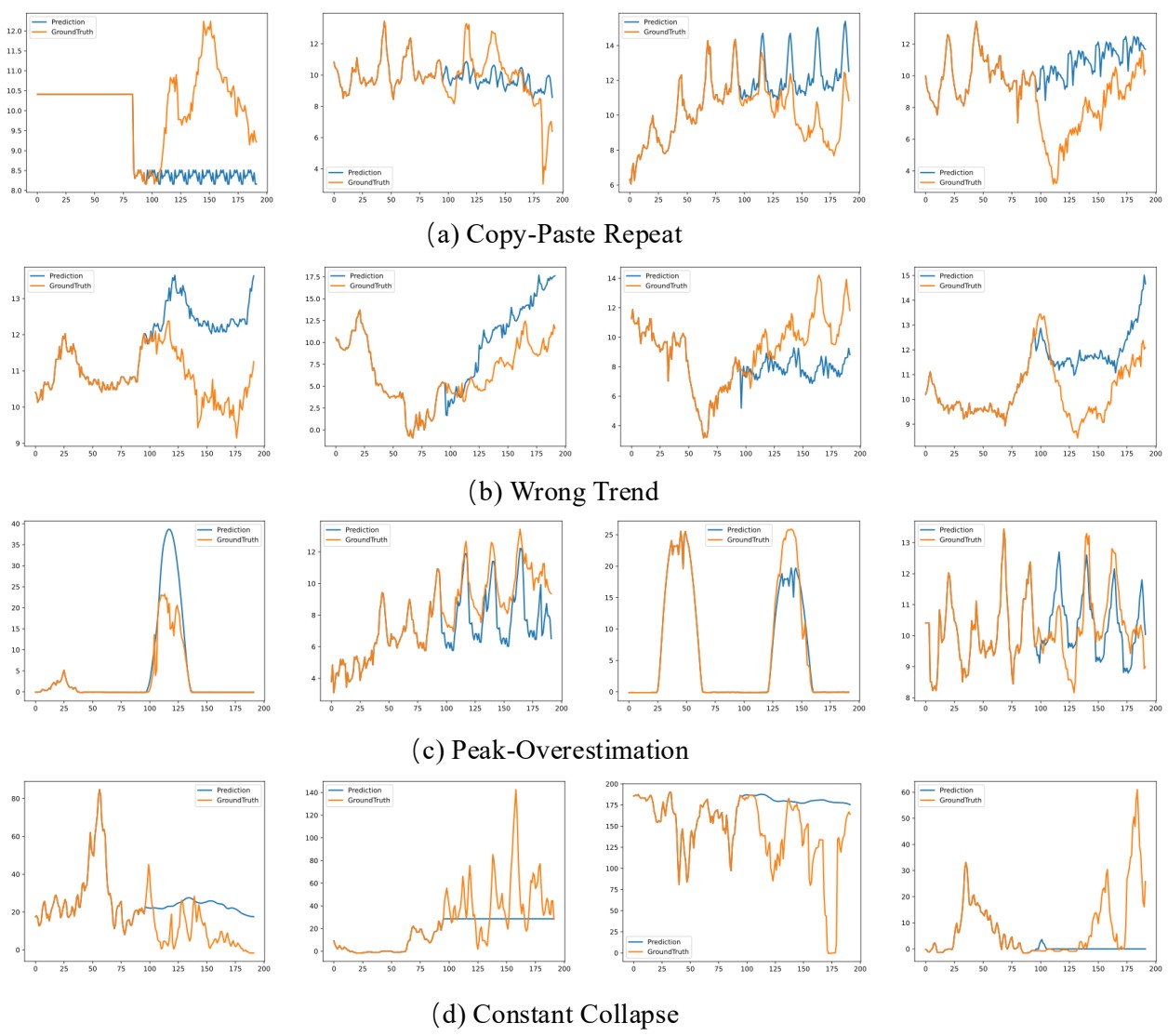

*Figure 8.* Visualization of typical failure modes: (a) Copy-Paste Repeat (naive history replication); (b) Wrong Trend (significant trajectory divergence); (c) Peak-Overestimation (amplitude exaggeration); and (d) Constant Collapse (degeneration into uninformative flat lines).

- **TimeXer** (Wang et al., 2024): An advanced Transformer framework designed to effectively empower time series forecasting by incorporating and aligning exogenous variables.

- **ConvTimeNet** (Cheng et al., 2025b): A deep hierarchical fully convolutional network that captures multi-scale temporal patterns through adaptive segmentation and deformable patching.

- **LSTPrompt** (Liu et al., 2024a): An LLM-based method that decomposes time series into trend and seasonal components to construct specific prompts for guiding the language model's forecasting.

- **LLM-Time** (Gruver et al., 2023): A zero-shot method that treats time series forecasting as a next-token prediction task by directly tokenizing numerical data for pre-trained LLMs.

- **TimeReasoner**: An approach that leverages the reasoning capabilities of LLMs to infer temporal dynamics and causal relationships within the time series data.

- **Time-LLM** (Jin et al., 2024): A comprehensive framework that aligns time series modalities with the text space of LLMs using reprogramming techniques and prompt-as-prefix strategies.

## A.3. Visualization

**Qualitative Analysis.** To intuitively evaluate the forecasting capability of our framework, we visualize the prediction results of MemCast alongside state-of-the-art baselines on a representative challenging case characterized by high non-stationarity and sharp fluctuations, as shown in Figure 7.

Observations indicate distinct performance gaps across different model families. Traditional methods exhibit severe limitations: ARIMA degenerates into a flat line, failing to model any temporal dynamics, while Prophet captures a completely incorrect linear trend that diverges significantly from the ground truth. Deep learning baselines (e.g., PatchTST, DLinear, iTransformer), generally suffer from the "over-smoothing" problem; while they capture the general trend, they consistently underestimate the magnitude of sharp peaks and valleys and exhibit noticeable temporal lag. Other LLM-based approaches reveal stability issues; for instance, Time-LLM introduces significant high-frequency noise, and LSTPrompt exhibits abrupt discontinuities and hallucinations.

In contrast, MemCast demonstrates superior robustness and precision. It successfully captures the evolving trend without the lag often seen in deep models and accurately reconstructs the amplitude of extreme fluctuations. This qualitative superiority validates that our memory-augmented reasoning mechanism effectively empowers the LLM to understand complex temporal patterns beyond simple pattern matching.

**Failure Case Analysis.** To provide a balanced and comprehensive evaluation, we visualize representative failure cases of MemCast in Figure 8, specifically focusing on the Oil Temperature forecasting task under out-of-distribution (OOD) covariates. While MemCast achieves superior performance on average, a detailed qualitative inspection reveals that it is not immune to reasoning errors when confronted with extreme distribution shifts or conflicting signals. We identify four distinct failure modes.

First, we observe **Copy-Paste Repeat**, where the model exhibits an over-reliance on retrieved memory, mechanically repeating historical patterns while ignoring immediate deviations in the input window. This suggests that strong retrieval relevance can occasionally suppress the model's adaptive reasoning. Second, the model may suffer from **Wrong Trend**, where the forecasted trajectory diverges inversely from the ground truth. This is likely attributed to conflicting signals between the retrieved context and noisy covariates, leading the LLM to hallucinate an incorrect directional shift at critical turning points. Third, regarding **Peak-Overestimation**, the model correctly identifies the timing of events (e.g., spikes) but drastically exaggerates their magnitude. This implies that while the semantic reasoning captures the occurrence of fluctuations, the precise numerical scaling requires further calibration. Finally, **Constant Collapse** represents a failure where the generation degrades into a flat line, occurring when high uncertainty triggers excessive smoothing or when the LLM fails to construct a coherent temporal narrative.

These qualitative results highlight that explicitly grounding reasoning in historical experience, while effective, still faces challenges in OOD settings. The observed failures indicate that future work should focus on enhancing the conflict resolution mechanism between memory and current context, as well as improving the physical plausibility constraints to mitigate such hallucinations.

## A.4. Detailed Prompt Construction

To effectively adapt the general reasoning capabilities of Large Language Models (LLMs) to the specialized task of time series forecasting, we design a comprehensive and structured prompting strategy. As illustrated in StrategyBox 1, our prompt is not merely a static input query but a dynamic, modular instruction set that bridges the modality gap between numerical time series data and textual logical reasoning. The prompt construction consists of four critical components, each serving a distinct role in guiding the model's generation process:

**Role Initialization and Task Definition.** The prompt begins by establishing a specialized persona ("expert time series forecaster") to activate the domain-specific latent knowledge of the LLM. It explicitly defines the forecasting horizon (e.g., $y_{t+1:t+24}$) and the available information scope, ensuring the model understands the input-output relationship and the boundaries of the task.

**Experience-Conditioned Memory Retrieval.** A core innovation of our framework is the injection of In-Context Memory. Unlike standard few-shot prompting that only provides positive examples, our retrieval module explicitly incorporates "Failure Modes" (e.g., Covariate-Miscalibration) and derived "Preventative Rules." By exposing the model to past mistakes

---

### Forecasting Prompt

**Role & Context:** You are an expert time series forecaster. The task is to predict Nord Pool electricity prices ($y_{t+1:t+24}$) given a 168-hour history ($\mathcal{H}$) and 24-hour future covariate forecasts ($\mathcal{F}$).

**In-Context Memory (Few-Shot Retrieval):** You are provided with retrieved historical examples containing distinct failure modes. *Example excerpt:* "**Case 2 (Bad):** [Covariate-Miscalibration] The model applied fixed linear coefficients without validating against historical sensitivity, causing a $+4.32$ error.

    **Preventative Rule:** Never apply absolute linear adjustments when deviations exceed 15% of the historical range." Use these distilled lessons to avoid repeating documented errors.

**Input Context Stream:** The prompt structures the input into four blocks:

    (1) **Historical Data:** A serialized sequence of timestamps, Target $OT$, and covariates (Grid Load, Wind Power);

    (2) **Future Covariates:** Known future values for Load and Wind to guide the forecast trend;

    (3) **Statistical Summary:** Computed meta-features including $\mu, \sigma$, trend slope, and lag-1 correlation (0.93);

    (4) **Visual Reasoning:** An automated analysis describing the current regime as "Oscillating trend with high volatility and pronounced intraday swings."

**Feedback & Constraints (Reflective Loop):** The previous attempt triggered a strict Quality Control (QC) error: *Boundary jump too large* ($|\hat{y}_{t+1} - y_t| = 4.74 > 2.06$). You are now under **Hard Constraints**: The forecast must strictly maintain continuity with the last observed value ($y_t = 50.53$) and strictly bounded within $[36.12, 65.29]$.

**Instruction:** Synthesize the retrieved lessons and visual insights. Prioritize autoregressive patterns over covariate signals if the regime is stable. Generate the final vector $\hat{Y} \in \mathbb{R}^{24}$ in the strict format `<answer>...</answer>`.

---

(e.g., applying absolute linear coefficients without validation), we enforce a "negative constraint" mechanism, effectively instructing the model on what not to do, thereby reducing hallucination and improving logical robustness.

**Multi-View Context Integration.** To enable holistic reasoning, the context aggregates data from multiple views:

- Serialized Data: Converts the raw numerical history and future covariates into a sequence accessible to the LLM.

- Statistical Summary: Provides computed meta-features (e.g., mean $\mu$, standard deviation $\sigma$, trend slope) to anchor the model's numerical sensitivity.

- Visual Reasoning: Includes a textual description of the series' morphological characteristics (e.g., "Oscillating trend," "Intraday swings"). This translates visual patterns into semantic descriptions, aiding the LLM in identifying regime shifts that are difficult to discern from raw numbers alone.

**Dynamic Feedback and Constraint Injection.** The final component represents the "Reflective Loop" of our framework. If a previous forecast violates physical constraints or fails Quality Control (QC) checks (e.g., a "Boundary jump too large"), the system dynamically appends specific error feedback and hard constraints (e.g., strict boundary ranges $[36.12, 65.29]$) to the prompt. This transforms the forecasting process from a one-shot generation into an iterative refinement, ensuring that the final output maintains physical plausibility and continuity.

