# OpenReview forum: "MemCast: Memory-Driven Time Series Forecasting with Experience-Conditioned Reasoning"
_ICML.cc/2026/Conference — ICML 2026 regular_

### Official Review · Reviewer_EG1R · 2026-03-10

**Soundness:** 3
**Presentation:** 4
**Significance:** 2
**Originality:** 2
**Overall Recommendation:** 3
**Confidence:** 3

**Summary:**

The authors propose MemCast, a framework that enhances time series forecasting (TSF) by reframing it as an experience-conditioned reasoning task for Large Language Models. By organizing training data into a hierarchical memory, consisting of historical patterns, reasoning wisdom, and general laws, the system retrieves relevant context to guide the LLM during inference. This approach allows the model to leverage past experience for accurate predictions without the need for additional parameter retraining.

**Compliance With Llm Reviewing Policy:**

Affirmed.

**Final Justification:**

The author indicates the proposed method is an alternative of RAG, which shows novelty in addressing the existing RAG problem with unconventional approach. Thus, I raised my score. I am expecting the paper to have more theoretical reasoning to help with the conclusion.

**Key Questions For Authors:**

1. What happens when the test data contains extreme anomalies or shifts that is different from the training-set memory?
2. Can you provide any formal analysis, error bounds, or convergence guarantees regarding the *Reflective Loop*?
3. How does the framework perform on datasets with entirely different dynamics?

**Limitations:**

yes

**Strengths And Weaknesses:**

Strengths:
1. The paper addresses the limitation that many existing LLM-based forcasters treat instances as isolated reasoning tasks without explicitly accumulating reusable experience.
1. The framework evaluates its performance against multiple basedlines across various real world datasets.

Weaknesses:
1.  Integrating large language models with external memory or retrieval-augmented generation is an increasingly standard paradigm. Applying this existing architectural concept to time series data feels incremental rather than a fundamental algorithmic breakthrough.
1. The proposed time-series memory is an overcustomization and doesn't generalize the time-series problem. The specific components designed for experience accumulation appear highly engineered for the presented datasets. This heavily tailored pipeline raises significant concerns that the architecture overfits to specific dataset dynamics and relies too much on semantic prompting rather than providing a generalized, robust solution for diverse time-series environments.
1.  There is no insight about the theoretical reasoning behind the framework. The paper relies heavily on empirical heuristics. However, it lacks rigorous mathematical foundations, formal error bounds, or theoretical guarantees to explain exactly why this experience-conditioned reasoning will reliably converge or consistently avoid hallucinations under severe distributional shifts.

---

> ### Author Rebuttal · Authors · 2026-03-31
>
> We thank the reviewer for the careful reading and constructive feedback. We are encouraged that the motivation and empirical performance are recognized. Below, we clarify several points and further position our contributions to address the concerns.
>
> ---
>
> ### R1. On Novelty: Beyond a direct extension of RAG
> We respectfully clarify that MemCast is fundamentally different from standard RAG, rather than a direct extension of it. Existing RAG-based approaches mainly retrieve raw instances or textual knowledge to augment inputs. In contrast, MemCast introduces **structured temporal experience abstraction**, where memory is organized into **patterns, wisdom, and laws**. This hierarchical design enables **experience-conditioned reasoning** rather than retrieval-conditioned generation. Instead of directly injecting retrieved information into the input context, MemCast uses retrieved experience to **guide the reasoning trajectory of the forecasting process itself**. Therefore, it shifts the paradigm from **retrieve-and-predict** to **experience-driven reasoning**.
>
> ---
>
> ### R2. On Generalization: Abstraction enables transferability beyond dataset-specific patterns
> We thank the reviewer for raising the important concern regarding generalization, particularly whether the constructed memory may be overly dependent on specific scenarios.
> We would like to clarify that this concern involves two levels: the **generality of the paradigm** and the **specificity of the experience content**. At the paradigm level, MemCast is general because it formulates time series forecasting（TSF） as a **memory-enhanced reasoning** process through experience accumulation, abstraction, and utilization. At the content level, the stored experience is naturally task- or domain-relevant. Importantly, such specialization does **not** imply overfitting: MemCast does not memorize raw instances, but instead abstracts experience into hierarchical forms. In this sense, generality and specialization are not contradictory—the former lies in the modeling framework, while the latter provides the domain knowledge needed for effective forecasting.
> To further address this concern, we conduct a cross-dataset transfer experiment, where memory built on **NP** is directly applied to **PJM** without adaptation. This suggests that MemCast captures **transferable temporal experience** beyond dataset-specific patterns, likely because higher-level abstractions reflect shared dynamics such as trend evolution and periodicity.
> | Setting | Memory Source | Test Set | MSE ↓ | MAE ↓ |
> |:--------|:--------------|:---------|:-----:|:-----:|
> | w/o Memory | — | PJM | 39.51 | 4.33 |
> | Pattern-only | NP | PJM | 38.74 | 4.24 |
> | Wisdom-only | NP | PJM | 35.96 | 3.97 |
> | Law-only | NP | PJM | 36.48 | 4.01 |
> | Full-memory | NP | PJM | 34.88 | 3.86 |
> | Full-memory | PJM | PJM | 30.24 | 3.58 |
>
> ---
>
> ### R3. On Theoretical Guarantees: A paradigm-oriented perspective
> We agree that theoretical analysis is important. However, this work primarily aims to introduce a **new modeling paradigm**—experience-driven reasoning for time series forecasting—rather than focusing on deriving strict theoretical guarantees.
> That said, MemCast is not purely heuristic. Its design is motivated by:
> - **temporal abstraction principles** (pattern → law),
> - **experience-based reasoning in sequential decision-making**, and
> - **iterative refinement via reflection**, which resembles policy improvement mechanisms.
> Similar to many recent paradigm-shifting works, our focus is on introducing a new modeling perspective, while theoretical analysis remains an important future direction. We will clarify these connections and provide more conceptual justification. Formal theoretical guarantees under distribution shift are an important direction for future work.
>
> ---
>
> ### R4. On Distribution Shift and Robustness: Abstraction and reflection enable robustness
> Regarding robustness under extreme distribution shift, we acknowledge that any memory-based approach may face challenges when historical experience becomes less relevant. However, MemCast mitigates this issue through:
> - **law-level abstraction**, which captures invariant temporal dynamics such as trend evolution and periodicity that are shared across datasets,
> - and a **reflection mechanism**, which allows the model to iteratively detect and correct mismatches when retrieved experience is misleading.
> This is consistent with our cross-dataset experiment, where higher-level memory shows stronger robustness under distribution shift. These results suggest that MemCast does not rely solely on dataset-specific patterns.
>
> ---
>
> Overall, we believe the main contribution of this work is to **reframe TSF as an experience-driven reasoning problem**, supported by structured memory abstraction and a novel inference paradigm. We will revise the paper to better highlight these contributions and clarify potential misunderstandings, and we hope this addresses the reviewer’s concerns.

---

> > ### Author Rebuttal · Reviewer_EG1R · 2026-04-05
> >
> > I acknowledge the authors response, and all questions are answered. The score is raised.

---

> > > ### Author Response · Authors · 2026-04-06
> > >
> > > Dear Reviewer,
> > >
> > > Thank you for reading our rebuttal and for your follow-up comments. We appreciate that our response helped address your concerns and clarify the paper.
> > >
> > > We also appreciate your updated assessment. We will incorporate the main clarifications from the rebuttal into the final version to further improve the presentation of the motivation, formulation, and contributions.
> > >
> > > Thank you again for your feedback.
> > >
> > > Best regards,
> > >
> > > The Authors

---

### Official Review · Reviewer_xipa · 2026-03-12

**Soundness:** 3
**Presentation:** 3
**Significance:** 3
**Originality:** 3
**Overall Recommendation:** 5
**Confidence:** 4

**Summary:**

This paper proposes MemCast, a time series forecasting framework based on hierarchical memory, which reformulates the prediction problem as an empirical conditional reasoning process. By constructing three types of memory modules, namely historical pattern, reasoning wisdom, and general law, and combining trajectory selection and dynamic confidence update mechanisms, it achieves experience accumulation and continuous evolution without parameter updates. Experimental results show that it achieves performance improvements on multiple benchmark datasets.

**Compliance With Llm Reviewing Policy:**

Affirmed.

**Final Justification:**

Thank you to the authors for their careful rebuttal. I am satisfied with the additional clarifications and discussions regarding the method details. The rebuttal has addressed my concerns and helped improve the clarity of the paper. I decide to increase my socres. Some of these points could be incorporated into the revision to further improve the clarity of the paper.

**Key Questions For Authors:**

1. Is the default value m for trajectory sampling consistent? Does different datasets require different values for m?

2. How is the threshold $\tau$ for separating successful and failed trajectories in reasoning wisdom set? Is it highly sensitive to the results?

3. Is the training memory infinitely large or finite?

**Limitations:**

No, the discussion of limitation part is missing in this paper.

**Strengths And Weaknesses:**

Strengths:

S1. This paper addresses the lack of experience accumulation and continuous evolution in existing LLM time series prediction methods by proposing a hierarchical memory structure with a clear overall framework.

S2. From problem motivation and method design to experimental verification, the chapters are clearly structured, and diagrams and case studies help readers understand the framework and core ideas.

S3. Systematic comparisons and ablation analyses were conducted on multiple datasets, showing that the method provides a stable improvement, supporting the paper's main conclusions.

&nbsp;

Weaknesses:

W1. The paper does not currently discuss the differences between this method and RAG or agentic LLM, especially in this problem setting.

W2. The paper does not explicitly state whether the default parameter m in trajectory sampling is set uniformly across all datasets or is adjusted based on the dataset.

W3. Regarding the threshold $\tau$ used to distinguish successful and failed trajectories in reasoning wisdom, there is currently a lack of explanation regarding its setting method and its sensitivity.

W4. The paper does not explicitly state whether memory continuously grows during training or whether there are capacity limitations.

---

> ### Author Rebuttal · Authors · 2026-03-31
>
> We sincerely appreciate the reviewer’s valuable feedback and encouraging assessment of the clarity, motivation, and empirical effectiveness of our framework. Below, we provide careful clarifications regarding the reviewer’s main concerns.
>
> ---
>
> ### R1. Differences from RAG and agentic LLMs
>
> We agree that the distinction between our framework and related paradigms such as **RAG** or **agentic LLMs** should be made more explicit, especially in the context of time series forecasting. Our method is related to these paradigms in spirit, but differs from them in both **objective** and **problem formulation**. Compared with **RAG**, our framework does not simply retrieve external context to supplement the prompt. Instead, it organizes training experience into a **hierarchical memory** consisting of historical patterns, reasoning wisdom, and general laws, and uses this memory to guide forecasting through structured experience-conditioned reasoning. Compared with **agentic LLMs**, our framework does not focus on autonomous multi-step tool use, planning, or open-ended task execution. Instead, it is designed for a more specific setting, namely **time series forecasting**, where the goal is to improve prediction by accumulating, organizing, and reusing forecasting experience in a structured way. In this sense, our framework can be viewed as an **experience-centered reasoning framework** for time series forecasting, rather than a direct application of standard RAG or a general agentic LLM pipeline. We will add a dedicated discussion in the revised version to clarify these differences more explicitly.
>
> ---
>
> ### R2. On the configuration of trajectory sampling \(m\)
>
> The default value of \(m\) is set consistently across datasets in our experiments. We also observe that the method is not highly sensitive to moderate changes in \(m\), and we will clarify both the default setting and the sensitivity trend more explicitly in the paper.
>
> ---
>
> ### R3. Threshold setting and sensitivity analysis for reasoning wisdom
>
> Thank you for pointing out the need for clarification. The threshold τ for separating successful and failed trajectories is selected based on validation performance. To further examine sensitivity, we additionally evaluate a small range around the default value:
>
> | Threshold setting | MSE ↓    |
> | ----------------- | -------- |
> | τ - 0.05  | 37.4     |
> | τ         | **36.9** |
> |τ + 0.05   | 37.2     |
>
> The results remain stable within a reasonable range, suggesting that the method is not highly sensitive to the exact threshold choice. We will add this clarification to improve reproducibility.
>
> ---
>
> ### R4. On memory size
>
> The memory is not allowed to grow indefinitely. In practice, memory is maintained with bounded size through filtering and updating mechanisms, while dynamic confidence adaptation updates the weights of existing entries rather than unconditionally expanding memory.
>
> ---
>
> We would like to express our sincere gratitude for the reviewer’s thoughtful and helpful feedback. These comments are very helpful in improving the clarity, completeness, and overall presentation of our work. We hope our responses have clarified the main concerns, and we would greatly appreciate any further questions or suggestions.

---

> > ### Author Rebuttal · Reviewer_xipa · 2026-04-02
> >
> > Thank you to the authors for their careful rebuttal. I am satisfied with the additional clarifications and discussions regarding the method details. The rebuttal has addressed my concerns and helped improve the clarity of the paper. I decide to increase my socres. Some of these points could be incorporated into the revision to further improve the clarity of the paper.

---

> > > ### Author Response · Authors · 2026-04-02
> > >
> > > Dear Reviewer,
> > >
> > > Thank you very much for your positive feedback and for your recognition of the originality of our work.
> > >
> > > We are glad to know that our rebuttal has addressed your concerns. We also sincerely appreciate your suggestion to incorporate these clarifications and discussions into the revised version. We will carefully revise the manuscript to improve its clarity and overall presentation further.
> > >
> > > Thank you again for your valuable time and support.
> > >
> > > Best regards,
> > > The Authors

---

### Official Review · Reviewer_p3qy · 2026-03-13

**Soundness:** 3
**Presentation:** 4
**Significance:** 3
**Originality:** 4
**Overall Recommendation:** 5
**Confidence:** 5

**Summary:**

This paper proposes MemCast, a learning-to-memory framework that reformulates time series forecasting as an experience-conditioned reasoning task. It constructs hierarchical memory from training data — comprising historical patterns, reasoning wisdom, and general laws — to guide LLM inference via ﻿history pattern retrieval, trajectory selection, and rule-based reflection, with dynamic confidence adaptation enabling continual memory evolution without test-set leakage. Experiments demonstrate the effectiveness of the method.

**Compliance With Llm Reviewing Policy:**

Affirmed.

**Final Justification:**

After reading the rebuttal, I believe the authors have adequately addressed my main concerns. The clarification on the experience-conditioned reasoning paradigm makes the contribution clearer. Overall, this is an interesting and promising work on experience-driven reasoning for time series forecasting, and I maintain my original score.

**Key Questions For Authors:**

1. Do historical patterns, reasoning wisdom, and general laws produce similar effects in certain scenarios?
2. Can experience-conditioned reasoning be given a more formal definition? What are the essential differences between it and traditional in-context learning?
3. Are these rules generated entirely automatically, or do they involve manual selection steps?
4. Can this method be transferred to other time series tasks (such as anomaly detection)?

**Limitations:**

Yes.

**Strengths And Weaknesses:**

Strengths:
1. The experiments cover different datasets across short- and long-term forecasting settings. The ablation studies clearly show the contribution of each memory component and the dynamic confidence adaptation strategy, providing a reasonable level of methodological validation.
2. The paper is well organized. Figure 2 and the case study clearly explain the framework, and the failure analysis is transparent and helpful.
3. Addressing the lack of experience accumulation and continual evolution in existing LLM forecasters is a practically meaningful contribution. The model-agnostic design also makes the framework potentially applicable to a wider range of models.
4. Reformulating TSF as experience-conditioned reasoning is an interesting idea, combining history-enhanced reasoning, wisdom-driven trajectory exploration, rule-based reflection, and dynamic confidence adaptation. The hierarchical memory design provides a new perspective compared with prior LLM-based forecasting approaches.
Weaknesses:
1. While a hierarchical design of historical pattern, reasoning wisdom, and general law has been proposed, the functional boundaries between these three can be further clarified to more clearly reflect their independent roles and complementary relationships.
2. The concept of "experience-conditioned reasoning" is relatively intuitive, but providing a more formal definition and a clearer comparison with traditional in-context learning would help strengthen its theoretical positioning.
3. More details could be added to the generation process of general law, such as whether the rules are generated entirely automatically through induction, to enhance the transparency and reproducibility of the method.
4. A brief discussion of the potential scalability of this framework in other time series tasks (such as anomaly detection) would further enhance the work's impact.

---

> ### Author Rebuttal · Authors · 2026-03-31
>
> We greatly appreciate the reviewer for the highly positive evaluation and insightful comments. We are greatly encouraged that the reviewer recognizes both the core idea and the practical value of our work. Below, we address the reviewer’s questions and suggestions one by one.
>
> ---
>
> ### R1. On the roles of different memory components
>
> These components operate at different levels: **historical patterns** capture instance-level dynamics, **reasoning wisdom** captures reusable reasoning strategies, and **general laws** provide global constraints. Together, they form a hierarchical abstraction from local observations to high-level guidance. We will further clarify their distinct yet complementary roles. As a concrete example, we construct the memory based on the **NP** dataset and evaluate on **PJM** using different memory components separately.
>
> | Setting      | Memory Source | Test Set | MSE ↓ | MAE ↓ |
> | ------------ | ------------- | -------- | ----: | ----: |
> | w/o Memory   | —             | PJM      | 39.51 |  4.33 |
> | Pattern-only | NP            | PJM      | 38.74 |  4.24 |
> | Wisdom-only  | NP            | PJM      | 35.96 |  3.97 |
> | Law-only     | NP            | PJM      | 36.48 |  4.01 |
>
> These results further support the distinct roles of different memory components. **Historical patterns** are more sensitive to local and dataset-specific dynamics, while **reasoning wisdom** and **general laws** transfer better across datasets, indicating that they capture more reusable reasoning strategies and higher-level constraints.
>
> ---
>
> ### R2. On the formal definition
>
> We define *experience-conditioned reasoning* as forecasting conditioned on **explicit, structured external memory**, i.e.,
> $\hat{y} = f(x, \mathcal{M})$, where $\mathcal{M}$ encodes distilled experience, including historical patterns, reasoning trajectories, and abstract rules. This differs fundamentally from in-context learning (ICL), which conditions on **raw, unstructured examples within the prompt** and relies on implicit pattern matching. In contrast, our formulation introduces **structured and reusable experience**, enabling case-based reasoning guided by inductive biases rather than surface-level analogy.
>
> ---
>
> ### R3. On the rule generation
>
> The general laws are automatically induced via LLM-based summarization of extracted features, without manual selection. This process enables the model to capture global regularities in a data-driven manner. We will clarify this for better reproducibility.
>
> ---
>
> ### R4. On the scalability
>
> We agree that a brief discussion of the scalability of our framework to other time series tasks would further strengthen the broader impact of the paper. Although our current experiments focus on time series forecasting, the proposed framework is not inherently limited to this task. The core idea of **experience-conditioned reasoning** is to augment model inference with structured external memory, which can in principle be extended to other time series tasks that also require historical reference, reasoning support, and adaptive decision-making. For example, in **time series anomaly detection**, historical patterns can provide references for normal or abnormal temporal behaviors, reasoning wisdom can support reusable decision strategies for distinguishing hard cases, and general laws can offer high-level constraints for validating anomaly decisions. This suggests that the hierarchical memory design has the potential to generalize beyond forecasting. We will incorporate a brief discussion of this broader applicability in the revised version to better highlight the potential impact of the framework across different time series tasks.
>
> ---
>
> We are truly grateful for the reviewer’s highly positive evaluation and constructive suggestions. These comments are very valuable to us and further strengthen our confidence in the significance of this work. We have carefully considered each point and would be happy to discuss any further questions.

---

> > ### Author Rebuttal · Reviewer_p3qy · 2026-04-03
> >
> > Thank you for the rebuttal. After carefully reading your responses, I feel that my previous concerns and questions have been adequately addressed. In particular, the clarification on the experience-conditioned reasoning paradigm and its role in time series forecasting makes the contribution clearer and more convincing. Overall, this is an interesting and promising work that explores experience-driven reasoning for time series forecasting. So I will maintain my original score.

---

> > > ### Author Response · Authors · 2026-04-03
> > >
> > > Dear Reviewer,
> > >
> > > Thank you very much for your careful reading of our rebuttal and for your positive feedback. We greatly appreciate your encouraging assessment of our work as an interesting and promising exploration of experience-driven reasoning for time series forecasting.
> > >
> > > We will further improve the final version by incorporating the key clarifications from the rebuttal to make the paper clearer and stronger.
> > >
> > > Thank you again for your time, thoughtful evaluation, and continued support.
> > >
> > > Best regards,
> > >
> > > The Authors

---

### Official Review · Reviewer_6KUk · 2026-03-13

**Soundness:** 3
**Presentation:** 2
**Significance:** 3
**Originality:** 3
**Overall Recommendation:** 4
**Confidence:** 2

**Summary:**

This paper frames time-series forecasting by LLMs as a memory-conditioned reasoning task, where historical observations/experiences are summarised into hierarchical memory artefacts, in order to properly condition reasoning about future predictions. The authors achieve strong performance compared to baselines and ablate to show the necessity of many components of their proposed architecture.

**Compliance With Llm Reviewing Policy:**

Affirmed.

**Final Justification:**

The authors' proposed framework is certainly useful, as it brings LLM based time-series forecasting one step closer to well known classical multi-scale approaches. The authors have provided a number of very useful comparisons and ablations, yielding a framework that many practitioners may find useful to adopt.

**Key Questions For Authors:**

Are the individual components somehow grounded/interpretable/predictive compared to other hierarchical TF approaches, beyond just ablation impact on overall performance? For instance, do the "laws" genuinely generalise more broadly across datasets than the "history patterns"? Do the updates at each level follow what you'd expect from non-stationarities in the world at different levels of the generative model, for e.g. can you somehow show that pattern updates capture local changes in the data?

**Strengths And Weaknesses:**

The paper captures something that is often missing in LLM formulations of forecasting  - explicit compression of historical observations into *compact hierarchical representations* (that the authors frame as patterns, wisdom and laws) that enable conditionally independent predictions about future states - rather than relying simply on training or in-context learning to implicitly do this compression. This makes the formulation comparable with many other TSF approaches with explicit latents, and is quite a desirable feature of their approach. The authors seem to get good results on benchmarks, and ablate many of the individual components of their proposal in detail.

However they use a strong model (GPT-5), and it is clear from their backbone experiments that this has a big effect on performance, compared to baselines that use older/weaker LLMs - so it is unclear how much of the improvement on baselines is due to architecture vs backbone. Something that would improve the paper's readability & confidence in the approach is to understand the bird's eye motivation for the author's proposed multi-component architecture, before diving into specifics - as it has many different moving parts, it's a bit hard to understand the design decisions that went into incorporating so many components.

---

> ### Author Rebuttal · Authors · 2026-03-31
>
> We sincerely thank the reviewer for the constructive and insightful feedback. We are encouraged that the reviewer recognizes the value of explicit experience abstraction and the empirical effectiveness of our approach. Below, we carefully address the reviewer’s questions and suggestions one by one.
>
> ---
>
> ### R1. Concern about the closed-loop evaluation setup for method effectiveness
>
> After carefully reading this comment, we realize that our current presentation may not have made the **evaluation protocol** sufficiently clear. We would like to clarify that **all LLM-based methods, including both the baselines and MemCast, use the same backbone model as the reasoning engine in our experiments**. Therefore, the comparison is conducted under a **fair and controlled setting**, and the improvements over the baselines should be attributed to the **proposed architecture** rather than differences in the backbone. This experimental design is intended precisely to provide a **closed-loop and fair validation** of MemCast’s effectiveness.
>
> ---
>
> ### R2. Discussion of  the interpretability and generalizability of hierarchical memory components
>
> We agree that the current presentation may make the design appear complex, and we will improve the high-level explanation. The three components are designed to capture different levels of abstraction:
>
> - **Historical patterns** capture instance-level temporal dynamics.
> - **Reasoning wisdom** captures reusable reasoning strategies across samples.
> - **General laws** provide global constraints that ensure consistency.
>
> This decomposition follows a hierarchical view of time-series generation, where local patterns, reasoning processes, and global constraints operate at different levels. We will make this motivation clearer before introducing the technical details. To further support this design, we conduct a small cross-dataset analysis.  Specifically, we construct memory on the **NP dataset and evaluate on PJM**:
>
> | Setting      | Memory Source | Test Set | MSE ↓ | MAE ↓ |
> | ------------ | ------------- | -------- | ----: | ----: |
> | w/o Memory   | —             | PJM      | 39.51 |  4.33 |
> | Pattern-only | NP            | PJM      | 38.74 |  4.24 |
> | Wisdom-only  | NP            | PJM      | 35.96 |  3.97 |
> | Law-only     | NP            | PJM      | 36.48 |  4.01 |
>
> We observe that general laws exhibit stronger transferability across datasets, while historical patterns are more dataset-specific.  This is consistent with the intended hierarchical design.
>
>
> ---
>
> ### R3. Interest in whether pattern updates capture local changes in the data
> We understand the reviewer’s main concern to be whether **pattern memory presented in text form can still clearly reflect the dynamic characteristics of the data**, especially **local changes**. We agree that this is an interesting question, because an effective hierarchical memory should improve overall performance, including enabling historical patterns to reflect changes in the data.
> In our framework, the **historical pattern** layer is designed to capture local temporal variations at the instance level, making it the most sensitive component to short-term trends, fluctuations, and seasonal changes. Specifically, when the framework is applied to time windows with different local dynamics, the retrieved patterns tend to produce distinct summaries that reflect these local changes, rather than collapsing them into a single global description.
> In addition, to avoid potential distribution leakage into the test set, we restrict memory updates to the weights of entries constructed and stored solely from the training data. We hope this clarification addresses your interest. If there are any further questions, we would be happy to discuss this point further.
>
> ---
>
> We sincerely thank the reviewer again for the positive evaluation and insightful suggestions. They also reinforce our confidence that this work explores a meaningful and promising direction.  We would also be very happy to engage in further discussion if any additional questions or concerns arise.

---

> > ### Author Rebuttal · Reviewer_6KUk · 2026-04-03
> >
> > Thanks to the authors for the detailed response!
> > - I appreciate the clarification about backbone, this substantially strengthens the results and I will amend my scores to reflect this
> > - The new bullet-point explanation of the three levels is extremely clear, and much more easy to understand- I second the decision to introduce this before diving into technical details.
> > - I like the experiment showing that general laws generalise more - interesting that wisdom (reusable strategies) generalises even more than laws - should I interpret this as trading off bias/variance, since laws are fixed/top-down/more likely to be biased, whereas wisdom can sometimes offer compositional generalisation, and sometimes not depending on the task distribution?
> > - re: the last question: can the authors provide some concrete evidence (even examples) of the following: "when the framework is applied to time windows with different local dynamics, the retrieved patterns tend to produce distinct summaries that reflect these local changes"... for instance, if I am looking at 3 successive time windows, where a true local change occured between 2 & 3, that the summaries from windows 1 & 2 are similar, but the summary for 3 is distinct, reflecting an underlying change?

---

> > > ### Author Response · Authors · 2026-04-03
> > >
> > > Dear Reviewer,
> > >
> > > Thank you again for the insightful follow-up questions and thoughtful interpretation of our results. We are glad that the clarifications were helpful.
> > >
> > > **(1) On patterns, wisdom, and laws.**
> > >  Your bias–variance interpretation is insightful. Empirically, we observe that laws are more stable but potentially biased, patterns are more local and data-specific, while wisdom (reasoning strategies) offers a flexible middle ground, enabling compositional generalization. This may explain why wisdom can sometimes generalize even better than laws.
> > >
> > > **(2) On local change (concrete example).**
> > >  We agree that concrete evidence is important. For example, in the ETTh dataset, considering three consecutive look-back windows ending at indices 387, 388, and 389: windows 387 and 388 exhibit similar dynamics and yield consistent pattern summaries, while 389 shows a local change and produces a distinct summary. This suggests that pattern memory captures local temporal variations and adapts to distributional changes. We will include a visualization in the revision for clarity.
> > >
> > > Thank you again for the constructive discussion. We believe these points help further clarify the role of hierarchical memory and its interaction with temporal dynamics, and we will incorporate these clarifications into the final version.
> > >
> > > Best regards,
> > >
> > > The Authors

---

### Decision · Program_Chairs · 2026-04-30

**Decision:**

Accept (regular)

**Comment:**

The paper proposes MemCast, a learning-to-memory framework that reformulates time series forecasting as an experience-conditioned reasoning task. It achieves this by organizing training data into a hierarchical memory structure consisting of historical patterns, reasoning wisdom, and general laws.

Three reviewers gave positive scores, explicitly stating in their final justifications and acknowledgments that the authors' rebuttal adequately addressed their main concerns regarding the backbone, baseline comparisons, and methodological boundaries. One reviewer maintained a Weak Reject score due to the lack of formal theoretical guarantees, but explicitly acknowledged the method's novelty compared to standard RAG in their final justification. As the paper's main contribution is to introduce an effective paradigm rather than having a theoretical derivation, and the empirical evidence is robust, the AC aligns with the majority consensus. Therefore, the overall recommendation is Accept.